# InfoOT: Information Maximizing Optimal Transport

## Abstract

Optimal transport aligns samples across distributions by minimizing the transportation cost between them, e.g., the geometric distances. Yet, it ignores coherence structure in the data such as clusters, does not handle outliers well, and cannot integrate new data points. To address these drawbacks, we propose InfoOT, an information-theoretic extension of optimal transport that maximizes the mutual information between domains while minimizing geometric distances. The resulting objective can still be formulated as a (generalized) optimal transport problem, and can be efficiently solved by projected gradient descent. This formulation yields a new projection method that is robust to outliers and generalizes to unseen samples. Empirically, InfoOT improves the quality of alignments across benchmarks in domain adaptation, cross-domain retrieval, and single-cell alignment.

## 1 Introduction

Optimal Transport (OT) provides a general framework with a strong theoretical foundation to compare probability distributions based on the geometry of their underlying spaces (Villani, 2009). Besides its fundamental role in mathematics, OT has increasingly received attention in machine learning due to its wide range of applications in domain adaptation (Courty et al., 2017; Redko et al., 2019; Xu et al., 2020), generative modeling (Arjovsky et al., 2017; Bousquet et al., 2017), representation learning (Ozair et al., 2019; Chuang et al., 2022), and generalization bounds (Chuang et al., 2021). The development of efficient algorithms (Cuturi, 2013; Peyré et al., 2016) has significantly accelerated the adoption of optimal transport in these applications.

Computationally, the discrete formulation of OT seeks a matrix, also called transportation plan, that minimizes the total geometric transportation cost between two sets of samples drawn from the source and target distributions. The transportation plan implicitly defines (soft) correspondences across these samples, but provides no mechanism to relate newly-drawn data points. Aligning these requires solving a new OT problem from scratch. This limits the applicability of OT, e.g., to streaming settings where the samples arrive in sequence, or very large datasets where we can only solve OT on a subset. In this case, the current solution cannot be used on future data. To overcome this fundamental constraint, a line of work proposes to directly estimate a mapping, the pushforward from source to target, that minimizes the transportation cost (Perrot et al., 2016; Seguy et al., 2017). Nevertheless, the resulting mapping is highly dependent on the complexity of the mapping function (Galanti et al., 2021).

OT could also yield alignments that ignore the intrinsic coherence structure of the data. In particular, by relying exclusively on pairwise geometric distances, two nearby source samples could be mapped to disparate target samples, as in Figure 1, which is undesirable in some settings. For instance, when applying OT for domain adaptation, source samples with the same class should ideally be mapped to similar target samples. To mitigate this, prior work has sought to impose structural priors on the OT objective, e.g., via submodular cost functions (Alvarez-Melis et al., 2018) or a Gromov-Wasserstein regularizer (Vayer et al., 2018b;a). However, these methods still suffer from sensitivity to outliers (Mukherjee et al., 2021) and imbalanced data (Hsu et al., 2015; Tan et al., 2020).

This work presents *Information Maximization Optimal Transport* (InfoOT), an information-theoretic extension of the optimal transport problem that generalizes the usual formulation by infusing it with *global* structure in form of mutual information. In particular, InfoOT seeks alignments that maximize mutual information, an information-theoretic measure of dependence, between domains. To

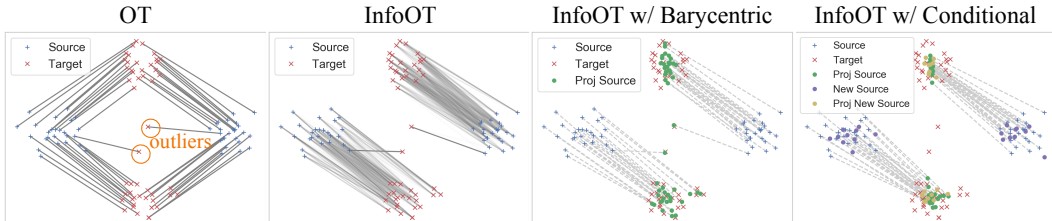

**Figure 1: Illustration of InfoOT on 2D point cloud.** Compared to classic OT, InfoOT preserves the cluster structure, where the source points from the same cluster are mapped to the same target cluster. For projection estimation (dashed lines), the new conditional projection improves over barycentric projection with better outlier robustness and out-of-sample generalization.

do so, we treat the pairs selected by the transportation plan as samples drawn from the joint distribution and estimate the mutual information with kernel density estimation based on the paired samples (Moon et al., 1995). Interestingly, this results in an OT problem where the cost is the log ratio between the estimated joint and marginal distributions $f_{XY}(x,y)/(f_X(x)f_Y(y))$. Empirically, we show that using a cost combining mutual information with geometric distances yields better alignments across different applications. Moreover, akin to Gromov-Wasserstein (Mémoli, 2011), the mutual information estimator only relies on intra-domain distances, which —unlike the standard OT formulation— makes it suitable for aligning distributions whose supports lie in different metric spaces, e.g., supports with different modalities or dimensionality (Alvarez-Melis & Fusi, 2020; Demetci et al., 2020).

By estimating a joint density, InfoOT naturally yields a novel method for out-of-sample transportation by taking an expectation over the estimated densities conditioned on the source samples, which we refer to as *conditional projection*. Typically, samples are mapped via a barycentric projection (Ferradans et al., 2014; Flamary et al., 2016), which corresponds to the weighted average of target samples, where the weights are determined by the transportation plan. The barycentric projection inherits the disadvantages of standard OT: sensitivity to outliers and failing to generalize to new samples. In contrast, our proposed conditional projection is robust to outliers and cross-domain class-imbalanced data (Figure 1 and 4) by averaging over samples with importance sampling, where the weight is, again, the ratio between the estimated joint and marginal densities. Furthermore, this projection is well-defined even for unseen samples, which widens the applicability of OT in streaming or large-scale settings where solving OT for the complete dataset is prohibitive.

In short, this work makes the following contributions:

- We propose InfoOT, an information-theoretic extension to the optimal transport that regularizes alignments by maximizing mutual information;
- We develop conditional projection, a new projection method for OT that is robust to outliers and class imbalance in data, and generalizes to new samples;
- We evaluate our approach via experiments in domain adaptation, cross-domain retrieval, and single-cell alignment.

## 2 RELATED WORKS

**Optimal Transport** Optimal transport provides an elegant framework to compare and align distributions. The discrete formulation, also called Earth Mover's Distance (EMD), finds an optimal coupling between empirical samples by solving a linear programming problem (Bonneel et al., 2011). To speed up the computation, Cuturi (2013) propose the Sinkhorn distance, an entropic regularized version of EMD that can be solved more efficiently via the Sinkhorn-Knopp algorithm (Knight, 2008). Compared to EMD, this regularized formulation typically yields denser transportation plans, where samples can be associated with multiple target points. Various extensions of OT have been proposed to impose stronger priors, e.g., Alvarez-Melis et al. (2018) incorporate additional structure by leveraging a submodular transportation cost, while Flamary et al. (2016) induce class coherence through a group-sparsity regularizer. The Gromov-Wasserstein (GW) distance (Mémoli, 2011) is a variant of OT in which the transportation cost is defined upon intra-domain pairwise distances. Therefore, GW has been adopted to align 'incomparable spaces' (Alvarez-Melis & Jaakkola, 2018; Demetci et al., 2020) as the source and target domains do not need to lie in the same space. Since the GW objective is no longer a linear program, it is typically optimized using projected gradient

descent (Peyré et al., 2016; Solomon et al., 2016). The Fused-GW, which combines the OT and GW objectives, was proposed by Vayer et al. (2018a) to measure graph distances.

**Mutual Information and OT** The proposed InfoOT extends the standard OT formulation by maximizing a kernel density estimated mutual information. Recent works (Bai et al., 2020; Khan & Zhang, 2022) also explore the connection between OT and information theory. Liu et al. (2021) consider a semi-supervised setting for estimating a variant of mutual information, where the unpaired samples are leveraged to minimize the estimation error. Ozair et al. (2019) replace the KL divergence in mutual information with Wasserstein distance and develop a loss function for representation learning. In comparison, the objective of InfoOT is to seek alignments that maximize the mutual information while being fully unsupervised by parameterizing the joint densities with the transportation plan. Another line of work also combines OT with kernel density estimation (Canas & Rosasco, 2012; Mokrov et al., 2021), but focuses on different applications.

## 3 BACKGROUND ON OT AND KDE

**Optimal Transport** Let $\{x_i\}_{i=1}^n \in \mathcal{X}^n$ and $\{y_i\}_{i=1}^m \in \mathcal{Y}^m$ be the empirical samples and $C \in \mathbb{R}^{n \times m}$ be the transportation cost for each pair, e.g,. Euclidean cost $C_{ij} = \|x_i - y_j\|$. Given two sets of weights over samples $\mathbf{p} \in \mathbb{R}_+^n$ and $\mathbf{q} \in \mathbb{R}_+^m$ where $\sum_{i=1}^n \mathbf{p}_i = \sum_{i=1}^m \mathbf{q}_i = 1$, and a cost matrix $C$, Kantorovich's formulation of optimal transport solves

$$\min_{\Gamma \in \Pi(\mathbf{p}, \mathbf{q})} \langle \Gamma, C \rangle, \quad \Pi(\mathbf{p}, \mathbf{q}) = \{\Gamma \in \mathbb{R}_+^{n \times m} | \gamma \mathbf{1}_m = \mathbf{p}, \gamma^T \mathbf{1}_n = \mathbf{q}\},$$

where $\Pi(\mathbf{p}, \mathbf{q})$ is a set of transportation plans that satisfies the flow constraint. In practice, the Sinkhorn distance (Cuturi, 2013), an entropic regularized version of OT, can be solved more efficiently via the Sinkhorn-Knopp algorithm. In particular, the Sinkhorn distance solves $\min_{\Gamma \in \Pi(\mathbf{p}, \mathbf{q})} \langle \Gamma, C \rangle - \epsilon H(\Gamma)$, where $H(\Gamma) = -\sum_{i,j} \Gamma_{ij} \log \Gamma_{ij}$ is the entropic regularizer that smooths the transportation plan.

**Kernel Density Estimation** Kernel Density Estimation (KDE) is a non-parametric density estimation method based on kernel smoothing (Parzen, 1962; Rosenblatt, 1956). Here, we consider a generalized KDE for metric spaces $(\mathcal{X}, d_\mathcal{X})$ and $(\mathcal{Y}, d_\mathcal{Y})$ (Li et al., 2020; Pelletier, 2005). In particular, given a paired dataset $\{x_i, y_i\}_{i=1}^n \in \{\mathcal{X}^n, \mathcal{Y}^n\}$ sampled i.i.d. from an unknown joint density $f_{XY}$ and a kernel function $K : \mathbb{R} \to \mathbb{R}$, KDE estimates the marginals and the joint density as

$$\hat{f}_X(x) = \frac{1}{n} \sum_i K_{h_1}\left(d_\mathcal{X}(x, x_i)\right); \quad \hat{f}_{XY}(x, y) = \frac{1}{n} \sum_i K_{h_1}\left(d_\mathcal{X}(x, x_i)\right) K_{h_2}\left(d_\mathcal{X}(y, y_i)\right), \quad (1)$$

where $K_h(t) = K(\frac{t}{h})/Z_h$ and the normalizing constant $Z_h$ makes equation 1 integrate to one. The bandwidth parameter $h$ controls the smoothness of the estimated densities. Figure 2 illustrates an example of KDE on 1D data. In this work, we do not need to estimate the normalizing constant as only the ratio between joint and marginal densities $\hat{f}_{XY}(x, y)/(\hat{f}_X(x)\hat{f}_Y(y))$ is considered while estimating the mutual information. For all the presented experiments, we adopt the Gaussian kernel:

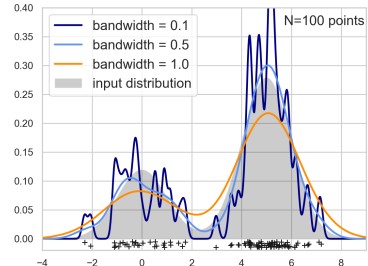

**Figure 2: Example of KDE.** The bandwidth affect the smoothness.

$$K_h\left(d_\mathcal{X}(x, x')\right) = \frac{1}{Z_h} \exp\left(-\frac{d_\mathcal{X}(x, x')^2}{2h^2\sigma^2}\right),$$

where $\sigma^2$ controls the variance. The Gaussian kernel has been successfully adopted for KDE beyond the Euclidean space (Li et al., 2020; Said et al., 2017), and we found it to work well in our experiments. For simplicity, we also set $h_1 = h_2 = h$ for all the experiments.

## 4 INFORMATION MAXIMIZING OT

Optimal transport captures the geometry of the underlying space through the ground metric in its objective. Additional information is not directly captured in this metric —such as coherence structure— will therefore be ignored when solving the problem. This is undesirable in applications

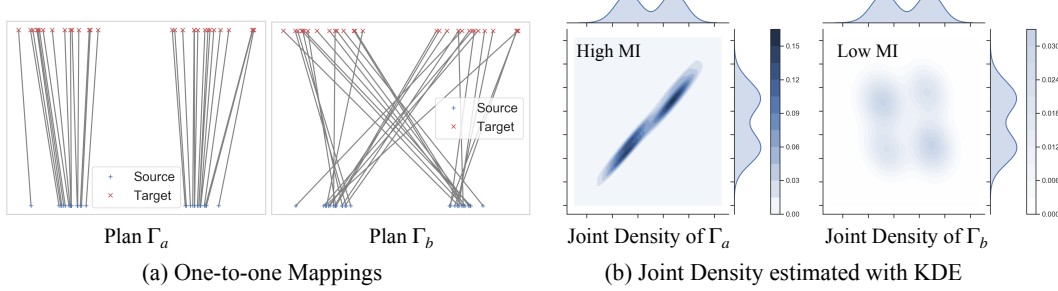

**Figure 3: Measuring Structure with Mutual information.** (a) The $\Gamma_a$ and $\Gamma_b$ are two one-to-one mappings where $\Gamma_a$ preserves the cluster structure and $\Gamma_b$ is a random permutation; (b) The estimated joint density of $\Gamma_a$ is more concentrated than the one of $\Gamma_b$, which also leads to higher mutual information under KDE.

where this additional structure matters, for instance in domain adaptation, where class coherence is crucial. As a concrete example, the cluster structure in the dataset in Figure 1 is ignored by classic OT. Intuitively, the reason for this issue is that the classic OT is too *local*: the transportation cost considers each sample separately, without respecting coherence across close-by samples. Next, we show that mutual information estimated with KDE can introduce global structure into OT maps.

## 4.1 Measuring Global Structure with Mutual Information

Formally, mutual information measures the statistical dependence of two random variables $X, Y$:

$$I(X, Y) = \int_{\mathcal{Y}} \int_{\mathcal{X}} f_{X,Y}(x, y) \log \left( \frac{f_{XY}(x, y)}{f_X(x) f_Y(y)} \right) dx dy \tag{2}$$

where $f_{XY}$ is joint density and $f_X, f_Y$ are marginal probability density functions. For paired datasets, various mutual information estimators have been defined (Belghazi et al., 2018; Moon et al., 1995; Poole et al., 2019). In contrast, we are interested in the inverse: given *unpaired* samples $\{x_i\}_{i=1}^n, \{y_j\}_{j=1}^m$, can we find alignments that maximize the mutual information?

**Discrete v.s. Continuous.** An immediate idea is to treat the discrete transportation plan $\Gamma$ as the joint distribution between $X$ and $Y$, and write the mutual information as $\sum_{i,j} \Gamma_{ij} \log(nm\Gamma_{ij}) = \log(nm) - H(\Gamma)$. In this case, maximizing mutual information would be equivalent to minimizing the entropic regularizer $H(\Gamma)$ introduced by (Cuturi, 2013). For a finite set of samples, this mutual information estimator is trivially maximized for *any* one-to-one mapping as then $H(\Gamma) = 0$. Figure 3 (a) illustrates two one-to-one mappings $\Gamma_a$ and $\Gamma_b$ between points sampled from multi-mode Gaussian distributions, where $\Gamma_a$ preserves the cluster structure and $\Gamma_b$ is simply a random permutation. They both maximize the mutual information estimate above, yet $\Gamma_a$ is a better alignment with high coherence. In short, directly using the transportation plan estimated from finite samples as the joint distribution to estimate mutual information between continuous random variables is problematic. In contrast, joint distributions estimated with KDE tend to be smoother, such as $\Gamma_a$ in Figure 3 (b). This suggests that KDE may lead to a better objective for the alignment problem.

## 4.2 InfoOT: Maximizing Mutual Information with KDE

Instead of directly interpreting the OT plan as the joint distribution for the mutual information, we use it to inform the definition of a different one. In particular, we treat $\Gamma_{ij}$ as *the weight of pair* $(x_i, y_j)$ *within the empirical samples drawn from the unknown joint distribution with density $f_{XY}$*. Intuitively, $\Gamma_{ij}$ defines what empirical samples we obtain by sampling from the joint distribution. Given a transportation plan $\Gamma$, the kernelized joint density in equation 1 can be rewritten as

$$\hat{f}_\Gamma(x, y) = \sum_i \sum_j \Gamma_{ij} K_h \left( d_{\mathcal{X}}(x, x_i) \right) K_h \left( d_{\mathcal{Y}}(y, y_j) \right). \tag{3}$$

The $1/n$ factor is dropped as the plan $\Gamma$ is already normalized ($\sum_{ij} \Gamma_{ij} = 1$). Specifically, we replace the prespecified paired samples in equation 1 with the ones selected by the plan $\Gamma$.

**Definition 1** (Kernelized Mutual Information). *The KDE estimated mutual information reads*

$$\hat{I}_\Gamma(X,Y) = \sum_{i,j} \Gamma_{ij} \log \frac{\hat{f}_\Gamma(x_i,y_j)}{\hat{f}(x_i)\hat{f}(y_j)} = \sum_{i,j} \Gamma_{ij} \log \frac{nm \cdot \sum_{k,l} \Gamma_{kl} K_h\left(d_\mathcal{X}(x_i,x_k)\right) K_h\left(d_\mathcal{Y}(y_j,y_l)\right)}{\sum_k K_h\left(d_\mathcal{X}(x_i,x_k)\right) \cdot \sum_l K_h\left(d_\mathcal{Y}(y_j,y_l)\right)}.$$

The estimation has two folds: (1) approximating the joint distribution with KDE, and (2) estimating the integral in equation 2 with paired empirical sample $(x_i, y_j)$ weighted by $\Gamma_{ij}$. The normalizing constant $Z_h$ in equation (1) cancels out while calculating the ratio between joint and marginal probability densities. To maximize the empirical mutual information $\hat{I}_\Gamma(X,Y)$, the plan has to map close-by points $i,k$ to close-by points $j,l$. Maximizing this information can be interpreted as an optimal transport problem:

$$\text{(InfoOT)} \qquad \max_{\Gamma \in \Pi(\mathbf{p},\mathbf{q})} \hat{I}_\Gamma(X,Y) = \min_{\Gamma \in \Pi(\mathbf{p},\mathbf{q})} \sum_{i,j} \Gamma_{ij} \cdot \log\left(\frac{\hat{f}(x_i)\hat{f}(y_j)}{\hat{f}_\Gamma(x_i,y_j)}\right). \qquad (4)$$

Instead of pairwise (Euclidean) distances, the transportation cost is now the log ratio between the estimated marginal and joint densities. The following lemma illustrates the asymptotic relation between the kernel estimated mutual information and the entropic regularizer.

**Lemma 2.** *When $h \to 0$ and $K(\cdot)$ is the Gaussian kernel, we have $\hat{I}_\Gamma(X,Y) \to -H(\Gamma) + \log(nm)$.*

When the bandwidth $h$ goes to zero, the estimated density is the sum of delta functions centered at the samples, and the estimated mutual information degenerates back to the standard entropic regularizer (Cuturi, 2013).

Note that the formulation of InfoOT does not require the support of $X$ and $Y$ to be comparable. Similar to Gromov-Wasserstein (Mémoli, 2011), InfoOT only relies on intra-domain distances, which makes it an appropriate objective for aligning distributions when the supports do not lie in the same metric space, e.g., supports with different modalities or dimensionalities, as section 6.4 shows.

**Fused InfoOT: Incorporating the Geometry.** When the geometry between domains is informative, the mutual information can act as a regularizer that refines the alignment. Along with a weighting parameter $\lambda$, we define the Fused InfoOT as

$$\text{(F-InfoOT)} \quad \min_{\Gamma \in \Pi(\mathbf{p},\mathbf{q})} \langle \Gamma, C \rangle - \lambda \hat{I}_\Gamma(X,Y) = \min_{\Gamma \in \Pi(\mathbf{p},\mathbf{q})} \sum_{i,j} \Gamma_{ij} \cdot \left(C_{ij} + \lambda \cdot \log\left(\frac{\hat{f}(x_i)\hat{f}(y_j)}{\hat{f}_\Gamma(x_i,y_j)}\right)\right).$$

The transportation cost becomes the weighted sum between the pairwise distances $C$ and the log ratio of joint and marginals densities. As Figure 1 illustrates, the mutual information regularizer excludes alignments that destroy the cluster structure while minimizing the pairwise distances. Practically, we found F-InfoOT suitable for general OT applications such as unsupervised domain adaptation (Flamary et al., 2016) and color transfer (Ferradans et al., 2014) where the geometry between source and target is informative.

### 4.3 NUMERICAL OPTIMIZATION

As the transportation cost is dependent on $\Gamma$, the objective is no longer linear in $\Gamma$ and cannot be solved with linear programming. Instead, we adopt the projected gradient descent introduced in (Peyré et al., 2016). In particular, Benamou et al. (2015) show that the projection can be done by simply solving the Sinkhorn distance (Cuturi, 2013) if the non-linear objective is augmented with the entropic regularizer $H(\Gamma)$. For instance, we can augment F-InfoOT as follows:

$$\min_{\Gamma \in \Pi(\mathbf{p},\mathbf{q})} \langle \Gamma, C \rangle - \lambda \hat{I}_\Gamma(X,Y) - \epsilon H(\Gamma).$$

In this case, the update of projected gradient descent reads

$$\Gamma_{t+1} \leftarrow \underset{\Gamma \in \Pi(\mathbf{p},\mathbf{q})}{\arg\min} \left\langle \Gamma, C - \lambda \nabla_\Gamma \hat{I}_{\Gamma_t}(X,Y) \right\rangle - \epsilon H(\Gamma). \qquad (5)$$

The update is done by solving the sinkhorn distance (Cuturi, 2013), where the cost function is the gradient to the objective of F-InfoOT. We provide a detailed derivation of (5) in Appendix A.2.

**Matrix Computation**    Practically, the optimization can be efficiently computed with matrix multiplications. The gradient with respect to the transportation plan $\Gamma$ is

$$\frac{\partial \hat{I}_\Gamma(X,Y)}{\partial \Gamma_{ij}} = \log\left(\frac{\hat{f}_\Gamma(x_i,y_j)}{\hat{f}(x_i)\hat{f}(y_j)}\right) + \sum_{k,l}\Gamma_{kl}\frac{K_h\left(d_\mathcal{X}(x_i,x_k)\right)K_h\left(d_\mathcal{Y}(y_j,y_l)\right)}{\hat{f}_\Gamma(x_k,y_l)}.$$

Let $K_X$ and $K_Y$ be the kernel matrices where $(K_X)_{ij} = K_h\left(d_\mathcal{X}(x_i - x_j)\right)$, $(K_Y)_{ij} = K_h\left((d_\mathcal{Y}(y_i - y_j)\right)$. The gradient has the following matrix form:

$$\nabla_\Gamma \hat{I}_\Gamma(X,Y) = \log\left(K_X\Gamma K_Y^T \oslash M_X M_Y^T\right) + K_X\left(\Gamma \oslash K_X\Gamma K_Y^T\right)K_Y^T$$

where $(M_X)_i = \hat{f}(x_i)$, $(M_Y)_i = \hat{f}(y_i)$ are the marginal density vectors and $\oslash$ denotes element-wise division. The gradient can be computed with matrix multiplications in $\mathcal{O}(n^2m + nm^2)$.

# 5    CONDITIONAL PROJECTION WITH INFOOT

Many applications of optimal transport involve mapping source points to a target domain. For instance, when applying OT for domain adaptation, the classifiers are trained on projected source samples that are mapped to the target domain. When $\mathcal{X} = \mathcal{Y}$, given a transportation plan $\Gamma$, a *barycentric projection* maps source samples to the target domain by minimizing the weighted cost to target samples (Flamary et al., 2016; Perrot et al., 2016). The mapping is equivalent to the weighted average of the target samples when the cost function is the squared Euclidean distance $c(x,y) = \|x - y\|^2$:

$$x_i \mapsto \underset{y \in \mathcal{Y}}{\arg\min}\sum_{j=1}^m \Gamma_{ij}\|y - y_j\|^2 = \frac{1}{\sum_{j=1}^m \Gamma_{ij}}\sum_{j=1}^m \Gamma_{ij}y_j. \tag{6}$$

Despite its simplicity, the barycentric projection fails when (a) aligning data with outliers, (b) imbalanced data, and (c) mapping new samples. For instance, if sample $x_i$ is mostly mapped to an outlier $y_j$, then its projection will be close to $y_j$. Similar problems occur when applying OT for domain adaptation. If the size of a same class differs between domains, false alignments would emerge due to the flow constraint of OT as Figure 4 illustrates, which worsen the subsequent projections.

Since the barycentric projection relies on the transportation plan to calculate the weights, any new source sample requires re-computing OT to obtain the transportation plan for it. This can be computationally prohibitive in large-scale settings. In the next section, we show that the densities estimated via InfoOT can be leveraged to compute the conditional expectation, which leads to a new mapping approach that is both robust and generalizable.

## 5.1    CONDITIONAL EXPECTATION VIA KDE

When treating the transportation plan as a probability mass function in the right-hand side of (equation 6), the barycentric projection resembles the conditional expectation $\mathbb{E}[Y|X = x]$. Indeed, the classic definition of barycentric projection (Ambrosio et al., 2005) is defined as the integral over the conditional distribution. But, this again faces the issues discussed in section 4. Instead, equipped with the densities estimated via KDE and InfoOT, the conditional expectation can be better estimated with classical Monte-Carlo importance sampling using samples from the marginal $P_Y$:

$$x \mapsto \underset{P_{Y|X=x}}{\mathbb{E}}[y] = \underset{y \sim P_Y}{\mathbb{E}}\left[\frac{f_{Y|X=x}(y)}{f_Y(y)}y\right] = \underset{y \sim P_Y}{\mathbb{E}}\left[\frac{f_{XY}(x,y)}{f_X(x)f_Y(y)}y\right] \approx \frac{1}{Z}\frac{\hat{f}_\Gamma(x,y_j)}{\hat{f}_X(x)\hat{f}_Y(y_j)}y_j \tag{7}$$

where $Z = \sum_{j=1}^m \hat{f}_\Gamma(x,y_j)/(\hat{f}_X(x)\hat{f}_Y(y_j))$ is the normalizing constant. Compared to the barycentric projection, the *importance weight* for each $y_j$ is the ratio between the joint and the marginal densities. To distinguish the KDE conditional expectation with barycentric projection, we will refer to the proposed mapping as *conditional projection*.

**Robustness against Noisy Data.**    By definition in equation 3, the joint density $\hat{f}_\Gamma(x,y)$ measures the similarity of $(x,y)$ to all other pairs selected by the transportation plan $\Gamma$. Even if $x$ is aligned with outliers or wrong clusters, as long as the points near $x$ are mostly aligned with the correct samples, the conditional projection will project $x$ to similar samples as they are upweighted by the joint density in (7). This makes the mapping much less sensitive to outliers and imbalanced datasets. See Figure 1 and Figure 4 for illustrations.

**Figure 4: Projection under imbalanced samples.** When the cluster sizes mismatch between source and target, barycentric projection wrongly projects samples to the incorrect cluster. In contrast, increasing the bandwidth of conditional projection gradually improves the robustness and yields better projection.

**Out-of-sample Mapping.** The conditional projection is well-defined for any $x \in \mathcal{X}$, and naturally generalizes to *new samples* without recomputing the OT. Importantly, the importance weight $\hat{f}_\Gamma(x,y)/(\hat{f}_X(x)\hat{f}_Y(y))$ can be interpreted as a similarity score between $(x,y)$, which is useful for retrieval tasks as section 6.3 shows.

The conditional projection tends to cluster points together with larger bandwidths that lead to more averaging. We found that using a smaller bandwidth (e.g., $h = 0.1$) for the conditional projection improves the diversity of the projection when the dataset is less noisy, e.g., the data in Figure 1. For noisy or imbalanced datasets, the same bandwidth used for optimizing InfoOT works well. Note that analogous to Lemma 2, when the bandwidth $h \to 0$, the conditional projection converges to the barycentric projection, making the barycentric projection a special case of the conditional projection (Figure 4).

## 6 EXPERIMENTS

We now evaluate InfoOT with experiments in point cloud matching, domain adaptation, cross-domain retrieval, and single-cell alignment. All the optimal transport approaches are implemented or adopted from the POT library (Flamary et al., 2021). Detailed experimental settings and additional experiments can be found in the appendix.

### 6.1 POINT CLOUD MATCHING

We begin with a 2D toy example, where both source and target samples are drawn from a Gaussian distribution with 2 modes, but the latter is rotated and has two outliers added to it, as Figure 1 shows. We compare the behavior of different variants of OT and mappings on this data. Perhaps not surprisingly, standard OT maps the source points in the same cluster to two different target clusters, overlooking the intrinsic structure of the data. In comparison, the alignment of InfoOT retains the cluster structure. On the right hand side, the barycentric projection maps two source points wrongly to the target outliers, while the conditional projection is not affected by the outliers. Lastly, we demonstrate an out-of-sample mapping with the conditional projection, where newly sampled points are correctly mapped to clusters.

Figure 4 depicts an class-imbalanced setting, where the corresponding clusters in source and target have different numbers of samples. Therefore, the barycentric projection wrongly maps samples from the same source cluster to different target clusters. When increasing the bandwidth in the conditional projection, the smoothing effect of KDE gradually corrects the mapping and yields more concentrated projections. In appendix B.1, we further demonstrate that InfoOT improves the baselines in a color transfer task, where pixels are treated as points in RGB space.

### 6.2 DOMAIN ADAPTATION

Next, we apply the fused version of InfoOT to two domain adaptation benchmarks: MNIST-USPS and the Office-Caltech dataset (Gong et al., 2012). The MNIST (M) and USPS (U) are digit classification datasets, and the Office-Caltech dataset contains 4 domains: Amazon (A), Dslr (D), Webcam (W) and Caltech10 (C), with images labeled as one of 10 classes. For MNIST and USPS, the raw images are directly used to compute the distances, while we adopt decaf6 features (Donahue et al., 2014) extracted from pretrained neural networks for Office-Caltech. Following previous works on OT for domain adaptation (Alvarez-Melis et al., 2018; Flamary et al., 2016; Perrot et al., 2016), the source samples are first mapped to the target, and 1-NN classifiers are then trained on the projected samples with source labels. The barycentric projection is adopted for all the baselines, while F-InfoOT is tested with both barycentric and conditional projection.

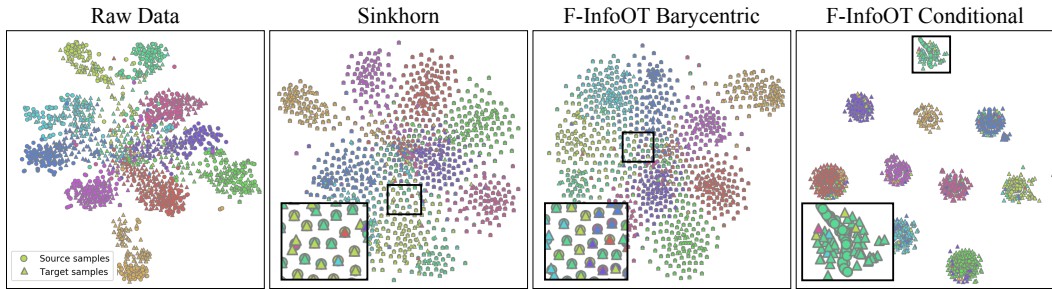

**Figure 5: tSNE visualization of projections.** We show the t-SNE visualization of projectrd source samples (circles) along with the target samples (triangles) on A→C. Classes are indicated by colors.

|  | OT | Sinkhorn | GL-OT | FGW | Linear | F-InfoOT | F-InfoOT* |
|---|---|---|---|---|---|---|---|
| **MNIST-USPS** | | | | | | | |
| M→U | 46.6±1.2 | 62.7±2.0 | 63.0±2.0 | 49.6±7.3 | 60.8±2.3 | **69.9±3.1** | 61.1±1.9 |
| U→M | 48.1±1.1 | 58.6±0.9 | 58.8±1.0 | 37.8±5.3 | 59.5±1.0 | **65.1±1.4** | 57.0±1.7 |
| **Office-Caltech** | | | | | | | |
| C→D | 61.3±10.9 | 71.9±15.9 | 76.9±18.6 | 61.3±10.9 | 58.1±12.5 | 78.1±13.9 | **87.5±7.8** |
| C→W | 64.0±7.0 | 66.0±9.3 | 66.7±9.4 | 64.0±7.0 | 62.0±7.4 | 79.7±4.3 | **81.0±6.7** |
| C→A | 78.6±4.7 | 77.3±4.8 | 83.3±6.1 | 79.0±4.5 | 77.9±6.7 | 87.0±3.7 | **90.6±2.0** |
| D→W | 90.7±3.8 | 90.7±4.7 | **93.7±4.3** | 90.7±3.8 | 89.0±6.3 | 91.0±4.1 | 93.3±4.7 |
| D→A | 73.8±4.5 | 73.4±2.9 | 84.4±2.9 | 73.8±4.5 | 72.7±3.9 | 83.6±4.3 | **89.8±1.8** |
| D→C | 67.0±3.0 | 66.4±3.8 | 76.5±2.9 | 67.0±3.0 | 67.5±4.1 | 70.2±3.7 | **80.8±1.8** |
| W→D | 81.3±7.8 | 80.6±10.4 | 85.6±10.6 | 81.3±7.8 | 79.4±7.2 | 79.4±8.9 | **89.4±11.8** |
| W→C | 64.8±4.6 | 65.8±4.1 | 73.5±5.4 | 64.8±4.6 | 65.5±4.5 | **75.8±3.3** | 74.4±3.7 |
| W→A | 67.3±4.9 | 69.3±5.4 | 79.6±3.0 | 67.3±4.9 | 70.5±4.8 | 85.5±1.9 | **89.3±2.3** |
| A→D | 73.1±10.6 | 68.8±8.8 | 76.3±8.2 | 73.1±10.6 | 66.9±7.8 | 80.6±7.5 | **81.3±9.8** |
| A→W | 64.7±6.3 | 70.0±7.5 | 70.0±7.0 | 64.7±6.3 | 69.3±6.6 | 82.0±7.2 | **87.0±4.6** |
| A→C | 65.4±5.3 | 69.3±6.0 | 79.8±5.8 | 65.5±5.7 | 66.9±4.5 | 74.4±3.4 | **81.2±3.6** |
| AVG | 71.0±2.5 | 72.4±3.7 | 78.9±4.5 | 71.0±2.5 | 70.5±2.4 | 80.6±3.3 | **85.6±3.3** |

**Table 1: Optimal Transport for Domain Adaptation.** The Fused-InfoOT with conditional projection (F-InfoOT*) performs significantly better than the barycentric counterpart (F-InfoOT) and the other baselines when the dataset exhibit class imbalance, e.g., Office-Caltech.

|  | **Office-Caltech** | | | **ImageCLEF** | | |
|---|---|---|---|---|---|---|
|  | P@1 | P@5 | P@15 | P@1 | P@5 | P@15 |
| L2-NN | 70.0±13.9 | 62.9±14.0 | 53.6±12.1 | 80.4±10.5 | 77.6±9.5 | 71.7±9.3 |
| Sinkhorn+NN | 69.0±11.6 | 65.1±6.4 | 58.0±6.9 | **81.9±10.2** | 81.1±10.6 | 79.2±10.4 |
| FGW+NN | 69.6±11.7 | 65.6±6.6 | 58.5±7.0 | 81.9±10.4 | 81.2±10.6 | 79.3±10.3 |
| F-InfoOT | **76.4±9.0** | **75.1±9.1** | **70.4±10.6** | 81.2±9.3 | **81.9±9.8** | **80.0±10.7** |

**Table 2: Optimal Transport for Cross-Domain Retrieval.** With conditional projection, InfoOT is capable to perform alignment for unseen samples without any modification.

Following Flamary et al. (2016), we present the results over 10 independent trials. In each trial of Office-Caltech, the target data is divided into 90%/10% train-test split, where OT and 1-NN classifiers are only computed on the training set. For MNIST-USPS, only 2000 samples from the source and target training set are used, while the original test sets are used. The strength of the entropy regularizer $\epsilon$ is set to 1 for every entropic regularized OT, and the $\lambda$ of F-InfoOT is set to 100 for all the experiments. The bandwidth for each benchmark is selected from $\{0.2, 0.3, ..., 0.8\}$ with the *circular validation procedure* (Bruzzone & Marconcini, 2009; Perrot et al., 2016; Zhong et al., 2010) on M→U and A→D, which is 0.4 and 0.5, respectively. We compare F-InfoOT with Sinkhorn distance (Cuturi, 2013), group-lasso regularized OT (Flamary et al., 2016), fused Gromov-Wasserstein (FGW) (Vayer et al., 2019), and linear-mapping OT (Perrot et al., 2016). For OTs involving intra-domain distances such as F-InfoOT and FGW, we adopt the following class-conditional distance for the source: $\|x_i - x_j\| + 5000 \cdot \mathbb{1}_{f(x_i) \neq f(x_j)}$, where the second term is a penalty on class mismatch (Alvarez-Melis & Fusi, 2020; Yurochkin et al., 2019) and $f$ is the labeling function. As Table 1 shows, F-InfoOT with barycentric projection outperforms the baselines in both benchmarks, demon-

strating that mutual information captures the intrinsic structure of the datasets. In Office-Caltech, many datasets exhibit the class-imbalance problem, which makes F-InfoOT with conditional projection significantly outperform the barycentric projection and the other baselines. Figure 5 visualizes the projected source and target samples with tSNE (Van der Maaten & Hinton, 2008). The barycentric projection tends to produce one-to-one alignments, which suffer from class-imbalanced data. In contrast, conditional projection yields concentrated projections that preserves the class structure.

| | **scGEM** | | **SNAREseq** | |
|---|---|---|---|---|
| | FOS, | Acc, | FOS. | Acc.. |
| UnionCom | 0.210 | 58.2 | 0.265 | 42.3 |
| MMD-MA | 0.201 | 58.8 | **0.150** | 94.2 |
| SCOT (GW) | 0.190 | 57.6 | **0.150** | 98.2 |
| InfoOT | **0.178** | **68.9** | 0.156 | **98.8** |

**Table 3: Single-Cell Alignment Performance.** Similar to GW, InfoOT also performs well in cross-domain alignment.

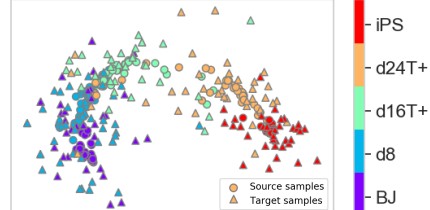

**Figure 6: scGEM alignment with InfoOT.** The cell types are indicated by colors.

### 6.3 CROSS-DOMAIN RETRIEVAL

We now consider unsupervised cross-domain image retrieval, where given a source sample, the algorithms have to determine the top-k similar target samples. Given fixed source and target samples, this can be formulated as an optimal transport problem, where the transportation plan $\Gamma_{ij}$ gives the similarity score between the candidate source sample $x_i$ and target samples $y_j$. Nevertheless, this formulation fails when new source samples come. For standard OT, one has to solve the OT problem again to obtain the alignment for new samples. In contrast, the importance weight $\hat{f}_\Gamma(x, y)/(\hat{f}_X(x)\hat{f}_Y(y))$ defined in conditional projection (7) naturally provides the similarity score between the candidate $x$ and each target sample $y$. We test F-InfoOT on the Office-Caltech (Gong et al., 2012) and ImageClef datasets (Caputo et al., 2014), where we adopt the same hyperparameter for Office-Caltech from the previous section. In the unsupervised setting, the in-domain transportation cost for the source is the Euclidean distance instead of the class-conditional distance. To compare with standard OTs, we adopt a nearest neighbor approach for the baselines: (1) retrieve the nearest source sample given an unseen sample, and (2) use the transportation plan of the nearest source sample to retrieve target samples. Along with a simple nearest neighbor retrieval baseline (L2-NN), the average top-k precision over 10 trials is shown in Table 2. The fuesed InfoOT significantly outperforms the baselines on Office-Caltech across different choices of $k$.

### 6.4 SINGLE CELL ALIGNMENT

Finally, we examine InfoOT in unsupervised alignment between incomparable spaces with the single-cell multi-omics dataset from (Demetci et al., 2020). Recent techniques allow to obtain different cellular features at the single-cell resolution (Buenrostro et al., 2015; Chen et al., 2019; Stoeckius et al., 2017). Nevertheless, different features are typically collected from different sets of cells, and aligning them is crucial for unified data analysis. We examine InfoOT with the sc-GEM (Cheow et al., 2016) and SNARE-seq (Chen et al., 2019) dataset provided by (Demetci et al., 2020) and follow the same data preprocessing steps, distance calculation, and evaluation setup. Here, two evaluation metrics are considered: "fraction of samples closer than the true match" (FOSCTTM) (Liu et al., 2019) and the label transfer accuracy (Cao et al., 2020). We compare InfoOT with UnionCom (Cao et al., 2020), MMD-MA (Liu et al., 2019), and SCOT (Demetci et al., 2020), where SCOT is an optimal transport baseline with Gromov-Wasserstein distance. Similarly, the bandwidth for each dataset is selected from $\{0.2, 0.3, ..., 0.8\}$ with the *circular validation procedure*. As Table 3 shows, InfoOT significantly improves the baselines on the sc-GEM dataset, while being comparable on the SNARE-seq dataset, demonstrating the applicability of InfoOT on cross-domain alignment. Figure 6 further visualizes the barycentric projection with InfoOT, where we can see that cells with the same type are well aligned.

## 7 CONCLUSION

In this work, we propose InfoOT, an information-theoretic extension of optimal transport. InfoOT produces smoother, coherent alignments by maximizing the mutual information estimated with KDE. InfoOT leads to a new mapping method, conditional projection, that is robust to class imbalance and generalizes to unseen samples. We extensively demonstrate the applicability of InfoOT across benchmarks in different modalities.

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

# A  PROOFS

## A.1  PROOF OF LEMMA 2

*Proof.* In the limit when $h \to 0$, the Gaussian kernel converges to

$$K_h(t) = \begin{cases} 1/Z_h & \text{if } t = 0 \\ 0 & \text{otherwise.} \end{cases}$$

Therefore, the kernel $K_h(d_\mathcal{X}(x_i, x_k))$ will only have non-zero value when $x_i = x_k$, which implies that the kernelized mutual information will converge as follows:

$$\begin{aligned}
\lim_{h \to 0} \hat{I}_\Gamma(X, Y) &= \lim_{h \to 0} \sum_{i,j} \Gamma_{ij} \log \frac{n^2 \cdot \sum_{k,l} \Gamma_{kl} K_h(d_\mathcal{X}(x_i, x_k)) K_h(d_\mathcal{Y}(y_j, y_l))}{\sum_k K_h(d_\mathcal{X}(x_i, x_k)) \cdot \sum_l K_h(d_\mathcal{Y}(y_j, y_l))}. \\
&= \sum_{i,j} \Gamma_{ij} \log \frac{n^2 \cdot \Gamma_{ij}/Z_h^2}{1/Z_h^2} \\
&= \sum_{i,j} \Gamma_{ij} \log \Gamma_{ij} + 2\log(n) \\
&= -H(\Gamma) + 2\log(n).
\end{aligned}$$

$\square$

## A.2  PROJECTED GRADIENT DESCENT

The classic mirror descent iteration is written as:

$$x_{t+1} \leftarrow \arg\min_x \{\tau \langle \nabla f(x_t), x \rangle + D(x\|x_t)\}.$$

When $D(y\|x)$ is the KL divergence: $D_{\mathrm{KL}}(y\|x) = \sum_i y_i \log \frac{y_i}{x_i}$, the update has the following form:

$$(x_{t+1})_i = e^{\log(x_t)_i - \tau \nabla f(x_t)} = (x_t)_i e^{-\tau \nabla f(x_t)}.$$

In our case, before the projection, the update reads

$$\Gamma'_{t+1} = \left(\Gamma_t \odot e^{-\tau(C - \lambda \nabla_{\Gamma_t} \hat{I}_{\Gamma_t}(X,Y) - \epsilon \nabla H(\Gamma_t))}\right).$$

Next, we solve the following projection w.r.t. KL metric:

$$\Gamma_{t+1} = \arg\min_{\Gamma \in \Pi(\mathbf{p}, \mathbf{q})} D_{\mathrm{KL}}(\Gamma\|\Gamma'_{t+1}).$$

As Benamou et al. (2015) shows, the KL projection is equivalent to solving the entropic regularized optimal transport problem, which is usually refer to the sinkhorn distance (Cuturi, 2013). Following (Peyré et al., 2016), we set the stepsize $\tau = 1/\epsilon$ to simplify the iterations and reach the following update rule:

$$\Gamma_{t+1} \leftarrow \arg\min_{\Gamma \in \Pi(\mathbf{p}, \mathbf{q})} \left\langle \Gamma, C - \lambda \nabla_\Gamma \hat{I}_{\Gamma_t}(X,Y) \right\rangle - \epsilon H(\Gamma).$$

# B  ADDITIONAL EXPERIMENTS

## B.1  COLOR TRANSFER

Color transfer aims to transfer the colors of the target images into the source image. Optimal transport achieves this by treating pixels as points in the RGB space, and maps the source pixels to the target ones. Here, 500 pixels are sampled from each image to compute the OT, then the barycentric projection is applied to map all the source pixels to target. We compare fused InfoOT with standard OT, Sinkhorn distance (Cuturi, 2013), and linear mapping estimation (Perrot et al., 2016) and show the results in Figure 7. We can see that InfoOT produces a sharper results than the baselines while decently recovering the colors in the target image.

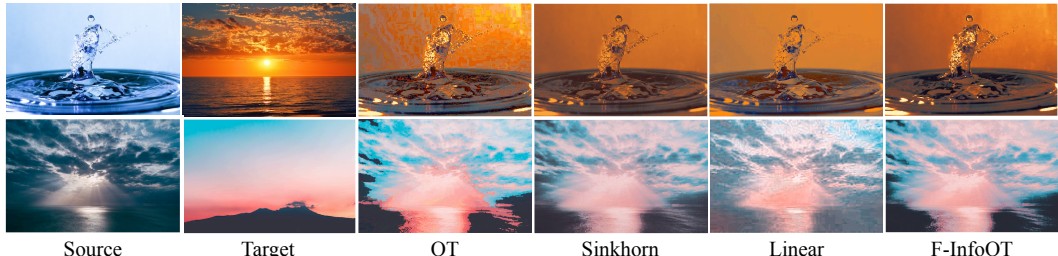

| Source | Target | OT | Sinkhorn | Linear | F-InfoOT |

**Figure 7: Color Transfer via Optimal Transport.** Fused InfoOT produces sharper results while preserving the target color compared to the baselines.

|  | | EN-ES | | EN-FR | | EN-DE | | EN-IT | | EN-RU | |
| --- | --- | --- | --- | --- | --- | --- | --- | --- | --- | --- | --- |
|  | Supervision | $\rightarrow$ | $\leftarrow$ | $\rightarrow$ | $\leftarrow$ | $\rightarrow$ | $\leftarrow$ | $\rightarrow$ | $\leftarrow$ | $\rightarrow$ | $\leftarrow$ |
| PROCRUSTES | 5K Words | 81.2 | 82.3 | 81.2 | 82.2 | 73.6 | 71.9 | 76.3 | 75.5 | 51.7 | 63.7 |
| Adv-NN | None | 81.7 | **83.3** | 82.3 | 82.1 | 74.0 | 72.2 | 77.4 | 76.1 | **52.4** | 61.4 |
| InvOT | None | 81.3 | 81.8 | 82.9 | 81.6 | 73.8 | 71.1 | 77.7 | 77.7 | 41.7 | 55.4 |
| InfoOT (h=0.55) | None | 81.6 | 78.5 | 82.4 | 80.5 | 75.4 | 74.2 | 78.6 | 75.7 | 48.1 | 52.9 |
| GW | None | **84.3** | 83.2 | **84.8** | **83.6** | **77.4** | **75.2** | **82.5** | **79.8** | 52.0 | **61.4** |

**Table 4: Cross-lingual Word Alignment.** The InfoOT achieves comparable performance to GW, demonstrating its potential in recovering cross-lingual correspondence.

### B.2 WORD EMBEDDING ALIGNMENT

Here, we explore the possibility of applying InfoOT for unsupervised word embedding alignment. We follow the setup in (Alvarez-Melis & Jaakkola, 2018), where the goal is to recover cross-lingual correspondences with word embedding in different languages. In this case, the pairwise distance between domains might not be meaningful, as the word embedding models are trained separately. Previous works suggest that cross-lingual word vector spaces are approximately isometric, which makes Gromov-Wasserstein an ideal choice due to its ability to align isometric spaces. Here, we treat GW as the oracle, and show that InfoOT can perform comparably to GW (Alvarez-Melis & Jaakkola, 2018) and other baselines such as InvOT (Alvarez-Melis et al., 2019), Adv-NN (Conneau et al., 2017), and supervised PROCRUSTES. We report the results on the dataset of Conneau et al. (2017) in Table 4, where both GW and InfoOT are trained with 12000 words and refined with Cross-Domain Similarity Scaling (CSLS) (Conneau et al., 2017). The entropy regularizer is 0.0001 and 0.02 for GW and InfoOT, respectively. We can see that InfoOT performs comparably with the baselines and GW, demonstrating its applicability in recovering cross-lingual correspondence.

### B.3 DIFFERENT HYPERPARAMETER FOR INFOOT

Here, we report the performance of InfoOT with different weights for entropic regularizer and mutual information on domain adaptation. As Table 6 shows, Fused-InfoOT performs consistently well across different hyperparameter selections.

### B.4 ADDITIONAL BASELINE: UNBALANCE OT

In this section, we additional include the results of unbalanced OT (UOT) Chizat et al. (2018); Frogner et al. (2015), which solves the following constrain optimization problem with generalized Sinkhorn-Knopp matrix scaling algorithm:

$$\min_{\Gamma \in \Pi(\mathbf{p}, \mathbf{q})} \langle \Gamma, C \rangle + \epsilon_m D_{\mathrm{KL}}(\Gamma \mathbb{1}, \mathbf{p}) + \epsilon_m D_{\mathrm{KL}}(\Gamma^T \mathbb{1}, \mathbf{q}) - \epsilon H(\Gamma).$$

We show the best results of UOT in Table 5 by selecting $\epsilon$ and $\epsilon_m$ within $(1, 5, 10)$ and $(0.5, 1, 10)$. We can see that InfoOT still outperform UOT by a non-trivial margin.

|  | F-InfoOT* | UOT |
| --- | --- | --- |
| C$\rightarrow$D | **87.5±7.8** | 81.9±11.9 |
| C$\rightarrow$W | **81.0±6.7** | 77.3±6.4 |
| C$\rightarrow$A | **90.6±2.0** | 87.8±3.5 |
| D$\rightarrow$W | **93.3±4.7** | 93.3±5.7 |
| D$\rightarrow$A | **89.8±1.9** | 87.8±3.2 |
| D$\rightarrow$C | **80.8±1.8** | 78.8±2.7 |
| W$\rightarrow$D | 89.4±11.8 | **98.8±2.6** |
| W$\rightarrow$C | 74.4±3.7 | **76.7±4.3** |
| W$\rightarrow$A | **89.3±2.3** | 80.1±3.3 |
| A$\rightarrow$D | **81.3±9.8** | 76.3±8.2 |
| A$\rightarrow$W | **87.0±4.6** | 69.3±8.6 |
| A$\rightarrow$C | **81.2±3.6** | 77.4±3.7 |
| AVG | **85.6±5.6** | 82.1±8.3 |

**Table 5: Unbalanced OT.**

| $(\lambda, \epsilon)$ | $(100, 1)$ | $(100, 10)$ | $(100, 20)$ | $(10, 1)$ | $(200, 1)$ |
|---|---|---|---|---|---|
| C→D | 87.5±7.8 | 86.3±8.7 | 85.0±8.9 | 87.5±7.8 | 86.9±8.0 |
| C→W | 81.0±6.7 | 86.7±7.7 | 88.0±5.9 | 80.0±6.1 | 81.7±7.2 |
| C→A | 90.6±2.0 | 90.5±2.1 | 90.7±2.1 | 89.4±2.4 | 90.6±2.0 |
| D→W | 93.3±4.7 | 92.7±6.0 | 91.3±5.3 | 93.3±5.7 | 94.0±4.4 |
| D→A | 89.8±1.9 | 89.9±2.1 | 89.6±1.9 | 89.6±1.7 | 89.8±1.5 |
| D→C | 80.8±1.8 | 81.2±1.9 | 81.5±1.6 | 80.7±1.8 | 80.6±1.8 |
| W→D | 89.4±11.8 | 86.9±10.4 | 83.8±11.5 | 91.9±12.2 | 90.0±11.9 |
| W→C | 74.4±3.7 | 74.2±3.7 | 74.0±3.4 | 74.2±4.6 | 74.4±3.8 |
| W→A | 89.3±2.3 | 89.3±2.0 | 89.3±2.0 | 86.4±2.8 | 89.3±2.3 |
| A→D | 81.3±9.8 | 80.6±9.5 | 82.5±11.7 | 81.9±10.4 | 82.5±9.2 |
| A→W | 87.0±4.6 | 83.3±6.3 | 83.0±6.0 | 83.8±6.5 | 87.0±4.6 |
| A→C | 81.2±3.6 | 80.8±4.0 | 80.2±3.9 | 81.2±3.3 | 82.2±2.7 |
| AVG | 85.6±5.6 | 85.2±5.3 | 84.9±5.1 | 85.0±5.6 | 85.7±5.5 |

**Table 6: InfoOT with different hyperparameters.** We test the Fused-InfoOT with conditional projection by varying the regularizer weights $(\lambda, \epsilon)$. Note that Table 1 in the main paper shows the results of $(\lambda = 100, \epsilon = 1)$.

| | 1-NN | 5-NN | 10-NN | 20-NN | Linear |
|---|---|---|---|---|---|
| OT | 71.0±8.8 | 77.0±6.4 | 78.3±4.8 | 77.5±6.8 | 77.8±8.6 |
| Sinkhorn | 72.4±7.3 | 76.0±5.3 | 76.3±4.0 | 75.2±6.3 | 76.7±9.8 |
| GL-OT | 78.9±7.3 | 80.7±5.3 | 80.5±4.3 | 78.2±7.1 | 78.1±9.7 |
| FGW | 71.0±8.8 | 76.9±6.5 | 78.3±4.8 | 77.5±6.8 | 77.5±7.9 |
| Linear | 70.5±8.4 | 75.9±6.2 | 77.4±5.4 | 77.5±7.1 | 76.7±7.5 |
| F-InfoOT | 80.6±5.7 | 81.4±5.8 | 79.7±5.0 | 76.4±7.1 | **82.9±7.0** |
| F-InfoOT* | **85.5±5.6** | **85.4±5.5** | **85.4±5.5** | **81.7±7.2** | 81.4±5.3 |

**Table 7: Results beyond 1-NN.** We evaluate the performance with $k$-NN classifiers and linear classifiers.

| | GFK | CORAL | SCA | JDA | TJM | DDC | DAN | MEDA | F-InfoOT* |
|---|---|---|---|---|---|---|---|---|---|
| C→D | 86.6 | 84.7 | 87.9 | 89.8 | 84.7 | 88.8 | 89.3 | 91.1 | 87.9 |
| C→W | 77.6 | 80.0 | 85.4 | 85.1 | 81.4 | 85.4 | 90.6 | 95.6 | 85.8 |
| C→A | 88.2 | 92.0 | 89.5 | 89.6 | 88.8 | 91.9 | 92.0 | 93.4 | 91.1 |
| D→W | 99.3 | 99.3 | 98.6 | 99.7 | 99.3 | 98.2 | 98.5 | 97.6 | 97.3 |
| D→A | 76.3 | 85.5 | 90.0 | 91.7 | 90.3 | 89.5 | 90.0 | 93.2 | 91.3 |
| D→C | 71.4 | 76.8 | 78.1 | 85.5 | 83.8 | 81.1 | 80.3 | 87.5 | 82.9 |
| W→D | 100 | 100 | 100 | 100 | 100 | 100 | 100 | 99.4 | 96.2 |
| W→C | 69.8 | 75.5 | 74.8 | 84.8 | 83.0 | 78.0 | 81.2 | 93.2 | 80.3 |
| W→A | 76.8 | 81.2 | 86.1 | 90.3 | 84.6 | 84.9 | 92.1 | 99.4 | 90.0 |
| A→D | 82.2 | 84.1 | 85.4 | 80.3 | 76.4 | 89.0 | 91.7 | 88.1 | 81.5 |
| A→W | 70.9 | 74.6 | 75.9 | 78.3 | 71.9 | 86.1 | 91.8 | 88.1 | 85.4 |
| A→C | 79.2 | 83.2 | 78.8 | 83.6 | 84.3 | 85.0 | 84.1 | 87.4 | 82.5 |
| AVG | 81.5 | 84.7 | 85.9 | 88.2 | 86.0 | 88.2 | 90.1 | 92.8 | 87.7 |

**Table 8: Baselines beyond OT.**

## B.5 Experiments beyond 1-NN Classifier

We report the performances of InfoOT and baselines with general $k$-NN classifiers and linear SVM classifiers in Table 7. We can see that fused-InfoOT consistently outperforms the baselines beyond 1-NN classifiers on Office-Caltech domain adaptation benchmark. In addition, compared to the baselines, the performance of InfoOT is more robust to the choice of the number of neighbors $k$.

## B.6 Baselines beyond Optimal Transport

We compare InfoOT with the following non-OT baselines: Geodesic Flow Kernel (GFK) (Gong et al., 2012), CORrelation Alignment (CORAL) (Sun et al., 2016), Scatter Component Analysis (SCA) (Ghifary et al., 2016), Joint distribution alignment (JDA) (Long et al., 2013), Transfer Joint

Matching (TJM) (Long et al., 2014a), Deep Domain Confusion (DDC) (Tzeng et al., 2014), Deep Adaptation Network (DAN) (Long et al., 2014b), and Manifold Embedded Distribution Alignment (MEDA) (Wang et al., 2018). For fair comparison, we report the performance of Fused-InfoOT calculated with full source and target dataset instead of the 10-fold setting in the main context. As Table 8 shows, InfoOT performs comparably to many baselines without training or finetuning neural networks.

## C  LIMITATIONS

While we have illustrated successful applications of InfoOT, there are limitations. One could expect InfoOT to perform worse when the geometry of input spaces provides little information. In particular, for raw inputs such as image datasets, InfoOT would not perform well without pre-extracted features. It is also non-trivial to directly apply InfoOT to very large-scale problems with millions of data points. Computational-efficient extensions such as mini-batch optimal transport (Nguyen et al., 2022) should be considered to apply InfoOT to large-scale datasets.

