# OpenReview forum: "InfoOT: Information Maximizing Optimal Transport"
_ICLR.cc/2023/Conference — Submitted to ICLR 2023_

### Official Review · Reviewer_EHDZ · 2022-10-22

**Confidence:** 3
**Correctness:** 3
**Technical Novelty And Significance:** 3
**Empirical Novelty And Significance:** 2
**Recommendation:** 6

**Clarity, Quality, Novelty And Reproducibility:**

- Can this be interpreted as an adaptive/dynamic cost function for OT? This is related to the weakness above. Overall, it seems odd and non-standard but may be an interesting idea that could be generalized.

- Could the KDE weighting function be used as a post-hoc smoothing method for standard OT?  It seems that it could be applied after doing empirical/entropic OT to provide smoother mappings.  If so, could you add it to the baseline methods?  This would help distinguish the mapping method from the OT training method.

- The discussion in Section 4.1 on discrete vs continuous may be premature.  It might be useful to discuss this later in the paper.

- Does equation 1 have a typo, it seems that the second equation should be $d_{\mathcal{Y}}$.

- Table captions should be on top of tables.


**Strength And Weaknesses:**

*Strengths*
- The proposed training method can maintain cluster structure when learning a mapping.

- The proposed mapping method can be more robust to outliers than barycentric proojection.

- The algorithm simplifies to a sequence of entropic OT problems.

- The results show improved performance compared to standard OT baselines across several tasks.

*Weaknesses*
- Kernel densities are known to breakdown in higher dimensions and suffer from the curse of dimensionality. This presents a key limitation and problem with the proposed approach. Why would this not be a problem here?  While there are experiments on MNIST for domain adaptation, it is unclear what is actually going on.

- The baselines for all external tasks seem to be OT-based. What about non-OT methods for these tasks?  For example, what about GAN-based methods for domain adaptation (e.g., [Ganin et al. 2016]) or Wasserstein dual formulations for learning OT maps via convex functions (e.g., [Makkuva et al., 2020])? There are also flow-based methods for alignment that could be used for domain adaptation [Grover et al. 2020, Usman et al., 2020]. I am unfamiliar with the sota methods for retrieval but it seems there would also be methods here. While I understand that you are comparing to other OT-based methods, the key motivation for this estimator seems to be for empirical reasons.  It is unclear that there is theoretic or "intrinsic" motivation for the proposed OT formulation. Thus, to demonstrate it's usefulness more broadly, it would be helpful to compare to sota methods for these tasks and/or do more intrinsic evaluation.

- Definition 1 seems odd or incorrect. It seems that Definition 1 is using two approximations that should be distinguished and explained. First, you approximate the joint distribution over x,y using a kernel density estimate.  If you were to use a plug-in estimator, this would still require an expectation using the kernel density for the expectation.  However, then you replace this expectation over the kernel density, with a psuedo-sample approximation that treats each possible pair of inputs (x,y) as a weighted empirical sample from this kernel density. Thus, the expectation turns into a weighted average with $\Gamma$ defininig the weights of each pairing.  I believe this is how you simplify to only using $\Gamma$ as the weights for the outer summation in Definition 1. Again, these two approximations should be discussed and distinguished as it was not clear from the original exposition.

- Using 1-NN in the source domain as the target predictor for domain adaptation and using KNN in the retrieval experiment are unlikely to do well in high dimensions and may be biased towards KDE-based non-parametric approaches. A standard CNN or fully-connected network could be used in evaluating domain adaptation. In particular, a KNN classifier may be unfairly biased towards KDE-like approaches as they are both non-parametric methods based on geometric distances.

- I'm not sure it is reasonable to have $\Gamma$ in the "cost" function of OT at least theoretically. It's unclear if it's actually OT anymore if $\Gamma$ is in the cost function itself.  Could  you explain why this is still OT if the cost function is now a function of the map itself?

Ganin, Y., Ustinova, E., Ajakan, H., Germain, P., Larochelle, H., Laviolette, F., ... & Lempitsky, V. (2016). Domain-adversarial training of neural networks. The journal of machine learning research, 17(1), 2096-2030.

Makkuva, A., Taghvaei, A., Oh, S., & Lee, J. (2020, November). Optimal transport mapping via input convex neural networks. In International Conference on Machine Learning (pp. 6672-6681). PMLR.

Grover, A., Chute, C., Shu, R., Cao, Z., & Ermon, S. (2020, April). Alignflow: Cycle consistent learning from multiple domains via normalizing flows. In Proceedings of the AAAI Conference on Artificial Intelligence (Vol. 34, No. 04, pp. 4028-4035).

Usman, B., Sud, A., Dufour, N., & Saenko, K. (2020). Log-likelihood ratio minimizing flows: Towards robust and quantifiable neural distribution alignment. Advances in Neural Information Processing Systems, 33, 21118-21129.


**Summary Of The Paper:**

The paper proposes a dynamic cost function for OT (see below on whether this is a valid OT objective or not) that attempts to maximize the mutual information of the projections using kernel density estimates---which depend on the coupling distribution.
This can be seen as a generalization of entropic regularization.
The InfoOT formulation can be merged with the standard OT problem and can then be solved via a sequence of entropic OT problems.
Empirically, this is shown to preserve cluster structure in 2D examples and qualitative t-SNE projections.
Furthermore, the paper proposes a mapping method based on the kernel density estimates called conditional projection that is more robust to projecting outliers.


**Summary Of The Review:**

Overall, I found the paper to address two interesting empirical problems related to using OT in ML: Preserving cluster structure and outliers. The technical descriptions were a bit vague but the overall idea of adding a mutual information regularizer seems interesting. However, I do have some concerns about the use of KDEs because they can break down in high dimensions, the experimental setup, and some technical points. I hope that at least some of these can be addressed by the author response.

---

> ### Author Response · Authors · 2022-11-11
> **Response to EHDZ**
>
> Thank you for your constructive and helpful suggestions. We would like to address your questions as follows:
>
> 1. Concerns toward KDE: The Gaussian kernel we used is based on the L2 distance between data points. We believe the L2 distance still provides meaningful information in a high-dimensional setting, which can be adopted to guide the optimization of InfoOT. In fact, the cost function of OT is also L2 distances.
>
> 2. Comparison to non-OT methods: We compare InfoOT with non-OT methods in Appendix B.6. As Table 8 shows, InfoOT performs comparably to many baselines without training or finetuning neural networks. The goal of InfoOT is not outperforming SOTA adaptation/retrieval methods, but serving as a new class of optimal transport that overcomes several critical limitations of OT.
>
> 3. Clarification of Definition 1. Thank you for pointing this out. Indeed, there are two approximations here. We have clarified this in the text below Definition 1.
>
> 4. Classifiers beyond 1-NN: As the baselines are also OT based approaches (non-parametric), we believe comparing 1-NN accuracy is fair under this setting. Another reason is that for office-caltech dataset, features extracted from pretrained deep neural networks (DeCAF6 feature) are adopted. This makes it less necessary to train another neural network based on it. To further validate InfoOT, we add additional results in Appendix B.5, where we include the results tested with 5-NN, 10-NN, 20-NN, and linear classifiers. We can see that fused-InfoOT consistently outperforms the baselines beyond 1-NN classifiers on the Office-Caltech domain adaptation benchmark.
>
> 5. Plan-dependent Cost Function: The Γ-dependent cost functions have appeared in several prior OT works. For instance, in the weak optimal transport formulation [1], the transportation cost is a function of Γ. In light of this, InfoOT can be interpreted as a special case of the weak OT formulation.
>
> To conclude, we want to highlight once again our main contributions. The formulation of InfoOT opens a new path to study the robustness and generalizability of optimal transport, which are two main challenges of optimal transport. It not only unifies the advantage from Monge / Kantorovich OT, Gromov-Wasserstein and unbalanced OT, but also improves the performance by a non-trivial margin. Thank you again for your suggestions, we hope that our clarifications help the reviewer in reassessing the paper.
>
> [1] Backhoff-Veraguas et al., Existence, duality, and cyclical monotonicity for weak transport costs

---

> > ### Comment · Reviewer_EHDZ · 2022-11-16
> > **Some concerns addressed but some remain**
> >
> > Hi authors,
> >
> > Thank you for the response. I appreciate the clarifications for definition 1 and plan-dependent cost functions. I also appreciate the additional results to confirm some things.
> >
> > However, I am still generally concerned about the KDE assumption. While L2 distance can be useful in high-dimensions, the density estimation of KDE is known to be poor. Your answer did not address this core problem.
> >
> > Additionally, perhaps the framing is the issue but it seems that the main reason to consider this formulation was motivated by practical reasons. Again, I'm still not sure that the problems are well-motivated from a theoretic perspective. Thus, the empirical results seem to be critical and it seems that they are only okay compared to other sota methods for these tasks. Thus, the practical benefit is still unclear.
> >
> > Nonetheless, I think this poses interesting ideas (albeit the presentation and clarity could be improved) so I still lean towards acceptance and will keep my current score.

---

### Official Review · Reviewer_pBFH · 2022-10-24

**Confidence:** 4
**Correctness:** 3
**Technical Novelty And Significance:** 3
**Empirical Novelty And Significance:** 3
**Recommendation:** 5

**Clarity, Quality, Novelty And Reproducibility:**

### Clarity. Very good.
I really appreciate the clarity of the presentation.

### Quality. Okay
This paper is well-written. Optimal transport is a versatile tool and there is growing interests in applying OT-based solutions to address various empirical problems. This work has exploited a simple smoothness heuristic to address the cluster preserving issue common to many OT applications. To make the points more intuitive the author(s) have presented visualization with both synthetic and real datasets.

Relevant literature is adequately cited. It will be nice to further complete the picture with refs on the more generic integral probability metric (IPM) and generalizations of OT such as Sinkhorn divergence. Also potentially compared with the more stable OT algorithms such as Inexact Proximal point method for Optimal Transport (IPOT), my hunch is that vanilla Sinkhorn is known to be unstable, and that might be the cause of issues discussed in the paper. Using more stable OT algorithms may address these pain points without any smoothness regularization.

### Novelty. Fair
KDE is a well-establish technique, and smoothness is a common regularity widely applied in machine learning.

### Reproducibility. Good
Details of the solution and experiment setups are very well documented, and there should be no major difficulty involved implementing the model.


**Strength And Weaknesses:**


### Strength
* Calling out the equivalence between entropy regularization and the proposed MI optimization makes an interesting point.
* The smoothness argument to preserve cluster structure makes good sense.
* Reported empirical gains on domain adaption and cross-domain retrieval are very encouraging

### Weakness
* As a criticism applies all kernel methods, the main concern is scalability of the method. First, the computation scales quadratically, and to ensure good performance, heavy parameter tuning is typically needed. Integration and end-to-end optimization with need neural nets (which is known to perform strongly even without the kernel trick), is often impractical with kernel-based components.
* There are existing work on the neural estimation of the transport plan and non-parametric estimation of MI (e.g., InfoNCE, NWJ, etc.). While integrating these two directions as a generalized version of the proposed solution seems to be out-of-scope, I would love to see some technical discussions at least.
* While I appreciate the the cluster-preserving idea, this is merely a flaky heuristic assumption and may not hold universally. Also some ablation study using alternative, more direct penalties to enforce smooth transport should be considered (e.g., the distance between nearby source points in the transported target space).
* For the domain adaption experiment, while I understand the author(s) followed the experiment setup from prior works, but 1-NN classification is usually considered unreliable. Numbers with alternative classification schemes should be reported (e.g., 5-NN, linear, simple neural net, etc.)
* There is no discussion on the limitations of the proposed method


**Summary Of The Paper:**

This paper proposed to improve the quality of optimal transport via simultaneously encouraging the mutual information (MI). This is justified by establishing the equivalence between the MI and the standard entropy regularization used in the Sinkhorn algorithm. Empirical results showed that when implemented with kernel density estimation, the proposed InfoOT better captures the cluster structure in data and works better on serval domain transfer tasks. An alternative view on the smoothness regularization is also given.  The author(s) further presented a conditional projection scheme for the OT mapping.

**Summary Of The Review:**

This paper is well-written. My major reservation is due to the lack of technical novelty and practical significance. KDE is a well-establish technique and it does not scale well, also for the problems considered in the experiments there should be better alternative solutions without applying OT (especially for the domain adaptation). I am open to reconsider my decision should more convincing argument or empirical results surface during the rebuttal phase.

---

> ### Author Response · Authors · 2022-11-11
> **Response to pBFH**
>
> Thank you for your constructive and helpful suggestions. We would like to address your questions as follows:
>
> 1. Computation Complexity: For source and target datasets of sizes n and m, respectively, the computation / time complexity of InfoOT is O(m*n^2 + m*n^2), which is the same as Gromov-Wasserstein. Empirically, for all the experiments, InfoOT experiments can reasonably converge within a few minutes. In addition, compared to standard OT such as Sinkhorn distance, there is only one additional bandwidth hyperparameter, which can be effectively selected via the unsupervised circular validation procedure as section 6.1 states.
>
> 2. Neural Estimation of MI:  It is indeed an exciting direction to integrate neural estimated MI with the current work. For instance, we can extend InfoOT with MINE [1] by solving mirror descent, where we alternatively optimize the transportation plan and the MI neural estimator. In this case, we will need to go through the whole optimization loop of MINE in each step, which could be computationally more expensive than our current formulation. Nevertheless, it is still an interesting direction to explore the combination between InfoOT and neural MI estimators.
>
> 3. Cluster Preservation: The cluster-preserving idea is indeed explored by some previous works, e.g., structured OT by Alvarez-Melis et al. Nevertheless, the formulation of InfoOT can do more than preserving cluster structure. In particular, the robustness and generalizability of InfoOT make it shine across different applications. Moreover, similar to Gromov-Wasserstein, InfoOT can align incomparable space.
>
> 4. Classifiers beyond 1-NN: We present the additional results in Appendix B.5. In particular, we additionally include the results of 5-NN, 10-NN, 20-NN, and a Linear SVM classifier in Table 7.  We can see that Fused-InfoOT consistently outperforms the baselines beyond 1-NN classifiers on the Office-Caltech domain adaptation benchmark. In addition, compared to the baselines, the performance of InfoOT is more robust to the choice of the number of neighbors k.
>
> 5. Limitations of InfoOT: We add a discussion of limitations in Appendix C.  While we have illustrated successful applications of InfoOT, there are limitations. One could expect InfoOT to perform worse when the geometry of input spaces provides little information. In particular, for raw inputs such as images, InfoOT would not perform well without pre-extracted features. It is also non-trivial to directly apply InfoOT to very large-scale problems with millions of data points. Computational-efficient extensions such as mini-batch optimal transport should be considered to apply InfoOT to large-scale datasets.
>
>
> To conclude, we want to highlight once again our main contributions. The formulation of InfoOT opens a new path to study the robustness and generalizability of optimal transports, which are two main challenges of optimal transport of optimal transport. It not only unifies the advantage from Monge / Kantorovich OT, Gromov-Wasserstein and unbalanced OT, but also improves the performance by a non-trivial margin. Thank you again for your suggestions, we hope that our clarifications help the reviewer in reassessing the paper.
>
> [1] Belghazi et al., MINE: Mutual Information Neural Estimation

---

### Official Review · Reviewer_BdbM · 2022-10-26

**Confidence:** 4
**Correctness:** 3
**Technical Novelty And Significance:** 3
**Empirical Novelty And Significance:** 3
**Recommendation:** 6

**Clarity, Quality, Novelty And Reproducibility:**

The proposed ideas are interesting. The proposed method, InfoOT can address several drawbacks from OT, especially about coherence structure (cluster, outlier) and the ability to integrate new data points.

I have some following concerns:
+ In case the input distributions are discrete, e.g., empirical distributions. It is not clear the advantages of using kernel density estimation (as in the proposed method) comparing to the entropic regularization (in Sinkhorn) for measuring global structure with mutual information (as in Section 4.1). Could the authors elaborate it with more details?

+ How many samples are required for the kernel density estimation (KDE)? and how to choose the bandwidth for the Gaussians used in KDE for the proposed InfoOT? especially for high-dimensional setting?

+ Could the author discuss the relation between the proposed InfoOT with Liu'2021 which is also based on mutual information and OT from given unpaired data?

+ For the robustness against noisy data, it is better if the authors compare the proposed method with the unbalanced OT approach (which also use to address this problem for OT). Could the authors discuss about it (and better to have some empirical comparison)?

+ In experiments, it is well-known that the entropic regularization affects performances of entropic OT, why the authors set it to 1 in experiments?
--- How the \lambda in Fused InfoOT affects its performances in applications? Why the authors set it to 100? Should one need to use \lambda to control the effect of the regularization?

**Strength And Weaknesses:**

Strength
+ The proposed InfoOT address several drawbacks of OT
+ The proposed method works well in applications.

Weaknesses
+ The advantage of the proposed method (using kernel density estimation for continuous setting) is not clear enough over discrete ones (e.g., entropic regularization in Sinkhorn), especially in the case input distributions are discrete (empirical distributions). It will be a plus if the authors elaborate this points with more details.
+ It is unclear how many samples are required for the kernel density estimation used in InfoOT

**Summary Of The Paper:**

The authors propose InfoOT which maximizes the mutual information between domains and minimize distances between input distributions. The proposed InfoOT address several drawbacks of OT, e.g., coherence structure (clustering, outliers) and easy to integrate new data points. Empirically, the authors evaluate the proposed method on domain adaption, cross-domain retrieval and single-cell alignment.

**Summary Of The Review:**

The proposed method is interesting. The proposed methods address several drawbacks of OT.

---

> ### Author Response · Authors · 2022-11-11
> **Response to BdbM**
>
> Thank you for your constructive and helpful suggestions. We would like to address your questions as follows:
>
> 1. Empirical Distribution and Discrete MI: The empirical distributions are still constructed based on samples drawn from continuous distributions. Whenever the underlying distribution is not discrete, it is problematic to estimate the densities with the discrete entropic regularizer used in Sinkhorn. In contrast,KDE is a well-known technique to approximate continuous distributions based on empirical samples. For instance, in Figure 3(a), the one-to-one alignments are constructed based on empirical samples drawn from 1D continuous Gaussian distributions, where their discrete entropic regularizers share the same value. Nevertheless, as Figure 3(b) shows, they have very different joint distribution and mutual information.
>
>
> 2. Sample Complexity and Bandwidth Selection: Our experiments span across different scales. For instance, there are 2533 datapoints in Office-Caltech (section 6.2), while there are only 177 samples in the scGEM dataset (section 6.4). The empirical results strongly support that InfoOT performs consistently across different sample sizes. In terms of bandwidth selection, we use the circular validation procedure as described in section 6.1. In particular, it is an unsupervised algorithm for selecting hyperparameters for unsupervised domain adaptation, and we found that it works well for selecting bandwidth in our setting. Moreover, it is quite robust in high-dimensional settings. We use raw images as input for MNIST-USPS, so the dimensionality is 28*28 = 784; the DeCAF feature used for Office-Caltech has 4096 dimensions.
>
> 3. Relation to Liu’2021: Liu et al. estimate the mutual information by leveraging weak supervision under a semi-supervised setting. Algorithmically, they approximate squared-loss mutual information by balancing the learning signal from a few paired data and unpaired data, where the objective can be solved with the Sinkhorn algorithm. In contrast, InfoOT optimizes standard mutual information without paired data, where the projected gradient descent is adopted to iteratively solve the optimization problem. In addition, the formulation of Liu et al. still suffers from sensitivity to noisy data and fails to generalize to unseen points.
>
> 4. Comparison to Unbalanced OT: Unbalanced OT (UOT) handles unbalanced distributions by relaxing the marginal constraints of the OT problem (more details in Appendix B.4). Therefore, one could also expect UOT improves the performance when facing imbalance data. Nevertheless, UOT does not take the data structure into account, and fail to generalize to new samples in its usual formulation. We empirically compare InfoOT and UOT in Appendix B.4. We can see that while UOT indeed improves the performance of vanilla OT, InfoOT still consistently performs better on domain adaptation benchmarks.
>
> 5. Hyperparameter of InfoOT: We set it to 1 for simplicity, as we would like to show that good empirical results can be achieved without extensive hyperparameter search. We also found that setting it to 1 works reasonably for all the OT approaches. Yes, the \lambda in fused OT is a standard balancing weighting hyperparameter for controlling the effect of regularization. We include more experimental results with different regularization strengths and choice of \lambda in Appendix B.3. As Table 6 shows, Fused-InfoOT performs consistently well across different hyperparameter selections.
>
> Thank you again for your suggestions, we hope that our clarifications help the reviewer in reassessing the paper.

---

> > ### Comment · Reviewer_BdbM · 2022-12-05
> > **Thank you for your explanation which partially addresses the raised concerns**
> >
> > Thank you for your explanation which partially addresses the raised concerns.
> > I also appreciate your effort to add experiments on the unbalanced OT in B4.
> >
> > The KDE component requires the underlying distribution is continuous.
> > Although InfoOT works well in experiments, the raised concerns about KDE (of InfoOT) still remain (e.g., for discrete measures, sample complexity for KDE in InfoOT).
> >
> > Overall, I think the proposed ideas are interesting for the community, some presentation and clarify could be improved (especially about KDE in InfoOT), I keep my score and lean on the acceptance.

---

### Decision · Program_Chairs · 2023-01-20

**Decision:**

Reject

**Justification For Why Not Higher Score:**

The paper is interesting. However, the relationship to the existing methods is still not clear. Moreover, the positive reviewers are not excited to publish at this point.

**Justification For Why Not Lower Score:**

N/A

**Metareview: Summary, Strengths And Weaknesses:**

In this paper, the authors propose a new optimal transport based on information maximization and name it infoOT. More specifically, the authors combine kernel density estimation (KDE) and optimal transport. The infoOT framework is interesting. However, the original idea of information maximization OT has existed since 2000s. For example, kernelized sorting (NIPS 2008) is highly related to the infoOT work. Moreover, as pointed by a reviewer, the comparison of Liu 21's work is also needed. Thus, I encourage the authors to update the paper based on the reviewer's comments and resubmit it to a future ML venue.






**Summary Of Ac-Reviewer Meeting:**

We discussed the paper carefully with the reviewers. However, the positive reviewers are not excited to publish at this point.